

# Multi-scale mining of kinematic distributions with wavelets

Ben G. Lillard[1*], Tilman Plehn[2†], Alexis Romero[1‡] and Tim M. P. Tait[1°]

**1** Department of Physics and Astronomy, University of California, Irvine, USA
**2** Institut für Theoretische Physik, Universität Heidelberg, Germany

* blillard@uci.edu, † plehn@uni-heidelberg.de, ‡ alexir2@uci.edu, ° ttait@uci.edu

## Abstract

Typical LHC analyses search for local features in kinematic distributions. Assumptions about anomalous patterns limit them to a relatively narrow subset of possible signals. Wavelets extract information from an entire distribution and decompose it at all scales, simultaneously searching for features over a wide range of scales. We propose a systematic wavelet analysis and show how bumps, bump-dip combinations, and oscillatory patterns are extracted. Our kinematic wavelet analysis kit KWAK provides a publicly available framework to analyze and visualize general distributions.



# 1 New Physics at Multiple Scales

Despite the proliferation of advanced statistical methods at the LHC, simple analyses of well-chosen kinematic distributions remain a powerful first attempt to tease out new physics with fuzzily specified characteristics. Resonances in invariant mass distributions or enhanced tails at high energies can reveal the existence of new particles produced on-shell, or the presence of heavy physics manifest as higher-dimensional operators, respectively.

Simple analyses are also particularly amenable to data-driven background determination. For example, a resonance search in an invariant mass distribution relies on a sideband fit, leading to a background-only hypothesis given as a simple functional form. At any point along the invariant mass distribution the analysis searches for an excess or bump via a sliding mass window. The underlying assumption is that the signal is a local excess, so the window is characterized by a scale related to the resonance width. This is also the origin of the look-elsewhere effect, which links the local significance to a global significance based on treating the entire distribution as one measurement.

The situation becomes more complicated when we search for more generic patterns. For example, quantum interference between the resonant signal and the smooth background typically implies that the deviation from the background becomes a deficit together with the excess, or a bump-dip [1–7]. It is particularly prominent when the resonant particle has a large width. A typical bump hunt combines the bump-dip to a net excess, considerably weakening the search.

There exist new physics models where modifications to the background are even less localized. Theories with compact extra dimensions [8,9] and their 4D product gauge group [10,11] or clockwork [12] analogues predict towers of states, implying periodic invariant mass patterns. While individual resonant structures are local and amenable to searches for bumps, an optimal search requires us to consider the entire distribution.

The general question for analyses of a single kinematic distributions is whether there exists an approach which balances the power of searching for local features with the flexibility of searches which retain information about longer scales or global features. Wavelet transforms are a standard tool which simultaneously decomposes data on an interval into different scales, allowing for sensitivity to local and global features. The wavelet transform

1. retains all information from the distribution in an orthogonal decomposition basis;
2. automatically zooms in to the proper resolution to match a given anomaly; and
3. retains all of the local information about the features of the distribution.

Wavelets have been successfully applied to a number of analyses in particle physics [13–19]. Applied to kinematic LHC data, they systematically evaluate the complete kinematic distribution, without any assumptions about the shape or scale of the potential anomaly. Because they represent an orthogonal change of basis, they maps the contents of a given number of bins onto the same number of wavelet coefficients, allowing us to mine a distribution for new physics without loss of information.

In this short paper we introduce the Haar wavelet transform as a tool to search for new physics in a kinematic LHC distribution. We introduce the Haar wavelet and illustrate its main features in Sec. 2, considering idealized deviations in the form of narrow and broad bumps, bump-dips, and an oscillatory pattern. In Sec. 3.1 we apply our analysis to simulated data inspired by the ATLAS di-photon invariant mass [20,21], injecting the same set of signal patterns. We analyze the actual ATLAS di-photon distribution in Sec. 3.2. Appendices include some details of the statistical analysis, and introduce our publicly available Python analysis package, KWAK.

## 2 Wavelet Transform

A Wavelet transform represents a given function in terms of simple orthonormal basis. In that sense it is similar to a Fourier transform, with the main difference that the wavelet basis retains a notion of locality in position space, which is relinquished by the Fourier transform.

### 2.1 Haar wavelet

A particularly simple wavelet is the Haar wavelet in one dimension [22], defined on the interval $x \in [0,1]$. The first two basis functions are

$$h_0(x) = 1 \qquad \text{and} \qquad h_1(x) = \begin{cases} +1 & x = 0 \dots 1/2 \\ -1 & x = 1/2 \dots 1 \,. \end{cases} \tag{1}$$

They characterize the over-all normalization of the function and its relative change from one side of the interval to the other, respectively. The next two basis functions are constructed from $h_1(x)$, compressed in $x$ by a factor of two,

$$h_{2,1}(x) = \sqrt{2}\, h_1(2x) \qquad\qquad h_{2,2}(x) = \sqrt{2}\, h_1(2x - 1) \,. \tag{2}$$

They characterize the change from one side of each subintervals to the other. Further basis functions continue to subdivide the intervals from the previous level. For example, the next step defines four functions, compressed by an additional factor two,

$$\begin{aligned} h_{3,1}(x) &= 2\, h_1(4x) & h_{3,2}(x) &= 2\, h_1(4x - 1) \\ h_{3,3}(x) &= 2\, h_1(4x - 2) & h_{3,4}(x) &= 2\, h_1(4x - 3) \,. \end{aligned} \tag{3}$$

Continuing to sub-divide the $x$-interval, the higher wavelet functions $h_{\ell,m}$ are organized in families labelled by level $\ell$ and increasingly localized in $x$. The label $m = 1 \dots 2^{\ell-1}$ specifies their position inside the interval. With the normalization $h_{\ell m} \propto 2^{(\ell-1)/2}$ the real wavelet functions are orthonormal,

$$\int_0^1 dx\, h_{\ell,m}(x)\, h_{\ell',m'}(x) = \delta_{\ell\ell'}\, \delta_{mm'} \,, \tag{4}$$

allowing the wavelet representation of a function $f(x)$ to be easily inverted,

$$f(x) = \sum_{\ell,m} \tilde{f}_{\ell,m}\, h_{\ell m}(x) \qquad \Longleftrightarrow \qquad \tilde{f}_{\ell,m} = \int_0^1 dx\, h_{\ell,m}(x) f(x) \,. \tag{5}$$

In this notation the similarity to a Fourier transform is manifest: the wavelets at each level resolve a waveform pattern that is the $\ell$th harmonic of the interval, but divided into $2^{\ell-1}$ locations along the interval, saturating the Nyquist criterion. The first coefficient $\tilde{f}_0$ is special in that it represents the over-all normalization of the distribution, and we will neglect it in most of our shape analysis below.

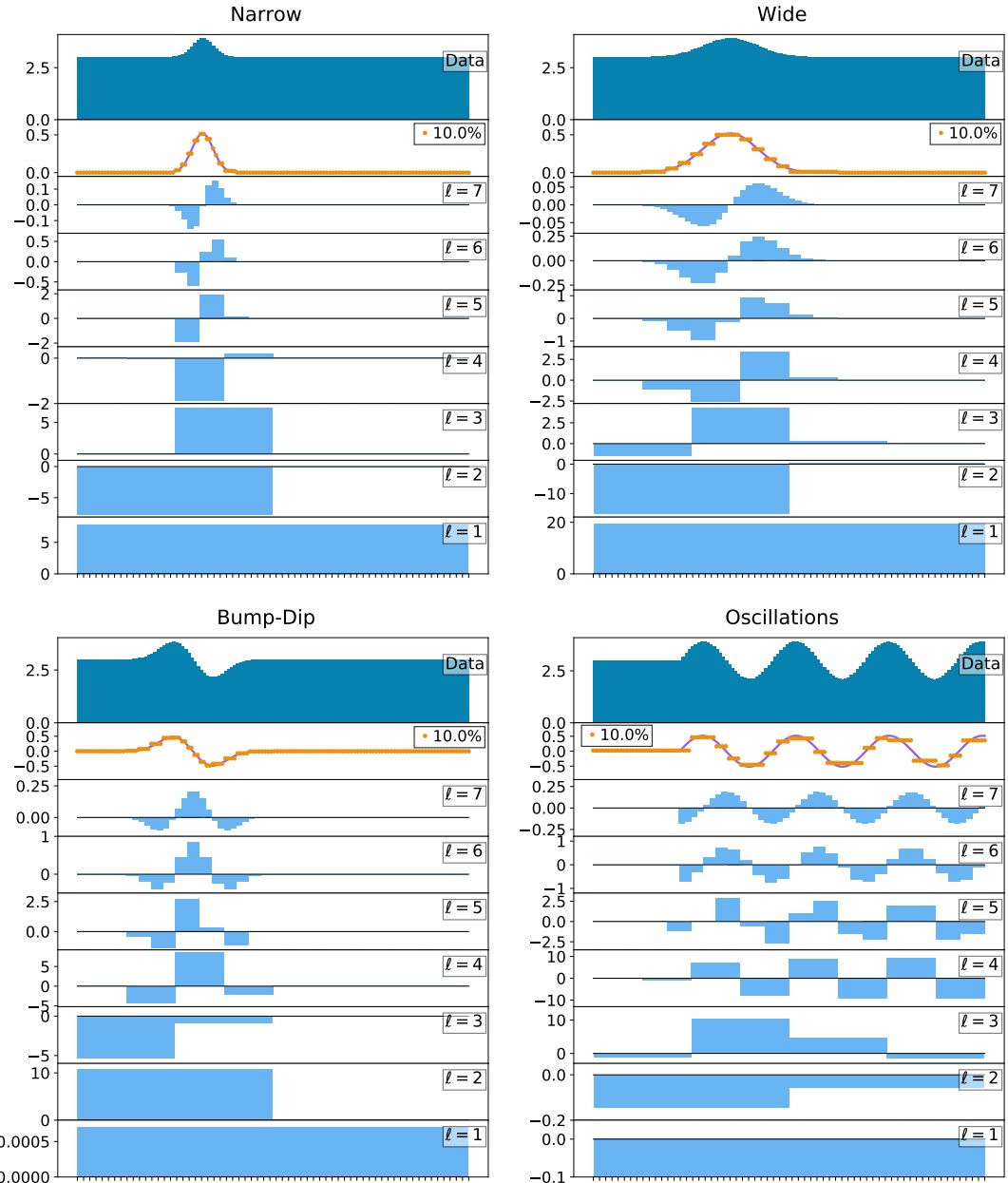

Figure 1: Toy wavelet analysis for a narrow (upper left) and a wide (upper right) bump, a bump-dip (lower left), and an oscillatory signal (lower right) on top of a flat background. The top panel shows the original distribution, the one below the pattern reconstructed retaining the largest 10% wavelet coefficients, and the remaining panels show the values of the wavelet coefficients $\tilde{f}_1$ through $\tilde{f}_{7,m}$ for 128 bins and no statistical fluctuations. The reconstructed signal (orange) is overlaid on top of the original (in purple). For each level the coefficients are aligned with their actual position in the distribution.

A kinematic distribution $f(x)$ with $2^L$ bins $f_j$ defines $L$ levels of wavelet coefficients. Including $\tilde{f}_0$, there are a total of $2^L$ wavelet coefficients, and the wavelet coefficients contain precisely the same information as the number of bin in the distribution. Because each wavelet basis state spans two distinct regions, the resolution at level $\ell$ corresponds to $2 \times 2^{\ell-1} = 2^\ell$ bins. From the definition of the wavelet transform in Eq.(5) it is clear that, for example, the

highest wavelet coefficients encode the $2^L/2$ pairwise differences between neighboring bins,

$$\tilde{f}_{L,m} = f_{2m-1} - f_{2m} \qquad \text{for } m = 1 \dots 2^{L-1}, \tag{6}$$

where in the discretized distribution $f(x) = f_j$ the bin index $j = 1 \dots 2^L$ replaces the continuous parameter $x$. The localized wavelet coefficients are aligned with the original distribution $f(x)$ such that at the highest level each wavelet coefficient $\tilde{f}_{L,m}$ corresponds to two bins $f_{2m-1}$ and $f_{2m}$, and the next level corresponds to four bins, etc. In many applications of the wavelet transformation it is standard to normalize the wavelet coefficients by a factor of $2^{(\ell-1)/2}$, but in our statistical analysis of integer-valued signals the definition in Eq.(6) is more convenient.

## 2.2 Toy Examples

In Fig. 1 we show the set of wavelet coefficients at each level for four toy distributions:

1. a narrow Gaussian bump;
2. a wide Gaussian bump;
3. a bump-dip combination; and
4. an oscillatory pattern with a shifted starting point.

Each distribution is added to a flat background and represented by a histogram with 128 bins. For the flat background alone all wavelet coefficients vanish by definition, Eq.(6). In each pane, the top panel shows the original histogram, and the lower panels show the wavelet coefficients from $\ell = 7$ to $\ell = 1$, followed by $\tilde{f}_0$ in the bottom panel. In this toy illustration we neglect statistical fluctuations, so the wavelet coefficients correspond perfectly to the source distribution. As discussed above, we align the wavelet coefficients of each level $\ell$ with the corresponding bins of the original distribution $f(x)$.

The upper left panel of Fig. 1 with the narrow bump illustrates how the large wavelet coefficients are localized at the position of the narrow excess. The largest wavelet coefficients appear at level $\ell = 5$, where the entire bump is covered by the two coefficients $\tilde{f}_{5,7}$ and $\tilde{f}_{5,8}$. This information encodes the fact that we are looking at a localized feature of size $1/2^5 \simeq 0.03$ of the original range $x = 0 \dots 1$. Interesting features can be reconstructed by considering a subset of the leading wavelet coefficients, which contain the most important information,

$$f_{\text{approx}}(x) = \sum_{\text{leading } \tilde{f}} \tilde{f}_{\ell,m} \, h_{\ell m}(x) \,. \tag{7}$$

By removing subleading coefficients, contributions of limited statistical significance are excised, allowing for sharp and robust image of the deviation from the background model. The second line in the upper left panel shows the result from the leading 10% of wavelet coefficients in size. Indeed, the small set of leading wavelets describe the bump pattern well, at the expense only of resolution from the highest level, $\ell = 7$. In the upper right panel we repeat this analysis for a bump with twice the width. As expected, most of the power is contained in the $\ell = 4$ coefficients.

The lower left panel of Fig. 1 describes a bump-dip, as it for example appears through quantum interference with wide resonances [6]. It is a challenge to the standard bump-hunting methods, which average the bump and the dip structures unless the resolution is sufficient and very carefully tuned. The total width of the feature is chosen to be about twice the width of the narrow bump, and indeed the largest wavelet coefficient is $\tilde{f}_{4,3}$, corresponding to the correct scale and position. At this scale, both the bump and the dip individually contribute positively to the wavelet coefficient.

Finally, an off-set oscillatory pattern is assumed for the lower right panel of Fig. 1. Such a modification poses a serious challenge for LHC searches [12]. The frequency of the pattern is such that most of its power appears at $\ell = 4$ with $m > 2$, reflecting the fact that the oscillations begin after an initial gap. We also show the approximate reconstructed signal, retaining the leading 10% wavelet coefficients, confirming that the signal pattern is again well described.

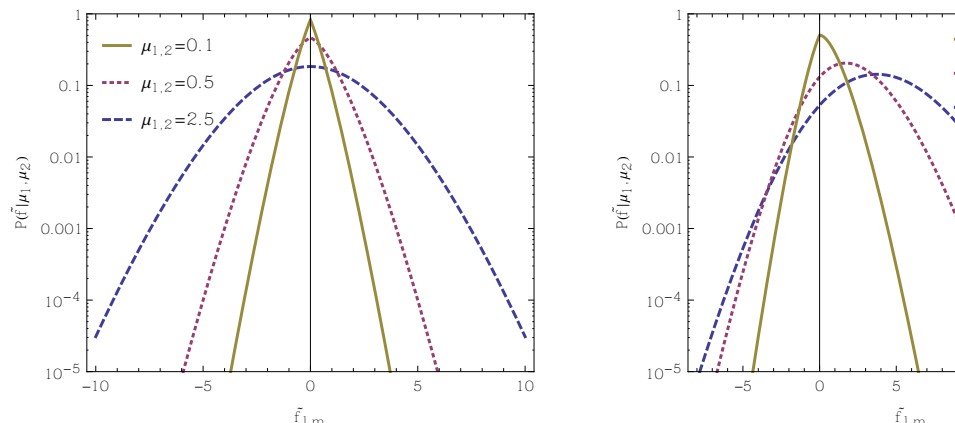

Figure 2: Statistical distribution for the wavelet coefficient $\tilde{f}$ assuming Poisson distributions of the two bins of the kinematic distribution $f_{1,2}$. The two input distributions are described by their means $\mu_{1,2}$.

## 2.3 Statistical Analysis

Realistic distributions inevitably contain statistical fluctuations. A kinematic distribution $f(x)$ is experimentally represented by $2^L$ bins $f_j$, where $f_j$ is the number of events in the $j$th bin and is integer-valued. If we assume that the bins are statistically independent, each bin count is described by a Poisson distribution with mean $\mu_j$,

$$P(f_j|\mu_j) = \frac{e^{-\mu_j}\,\mu_j^{f_j}}{f_j!}\,, \tag{8}$$

which implies that the probability distribution for the $m = 1$ wavelet coefficient of the highest level $\ell = L$ is

$$P(\tilde{f}|\mu_1,\mu_2) = \sum_{f_1,f_2} \left.\frac{e^{-\mu_1-\mu_2}\,\mu_1^{f_1}\mu_2^{f_2}}{f_1!f_2!}\right|_{\tilde{f}=f_1-f_2} = e^{-\mu_1-\mu_2}\left(\frac{\mu_1}{\mu_2}\right)^{\tilde{f}/2}\mathcal{I}_{\tilde{f}}(2\sqrt{\mu_1\mu_2})\,, \tag{9}$$

where $\mathcal{I}_n$ is the $n$th modified Bessel function of the first kind. This probability distribution is referred to as the Skellam distribution [23]. Its mean, variance, skew, and excess kurtosis are

$$\mu = \mu_1 - \mu_2\,, \qquad\qquad \sigma^2 = \mu_1 + \mu_2\,,$$
$$\gamma_1 = \frac{\mu_1-\mu_2}{(\mu_1+\mu_2)^{3/2}}\,, \qquad\qquad \gamma_2 = \frac{1}{\mu_1+\mu_2}\,. \tag{10}$$

When the Poisson distributions per bin in Eq.(8) becomes Gaussian, $\mu_1 + \mu_2 \gg 1$, $\gamma_1$ and $\gamma_2$ vanish, and $P(\tilde{f})$ approaches the expected Gaussian shape. We show the probability distribution for the wavelet coefficients in Fig. 2, assuming independent Poisson distributions for the bins of the underlying kinematic distribution. The tails of $P(\tilde{f})$ are exponentially suppressed, and as the mean values $\mu_{1,2}$ of the input distributions increase, the resulting $P(\tilde{f})$ indeed

approaches a Gaussian. In Appendix A, we provide the probability distribution $P(\tilde{f}|H_0)$ for generic values of $\ell \leq L$ and $m \geq 1$, and for a generic hypothesis pattern $H_0$.

A statistical analysis traces all of the correlations of the input distribution $f(x)$ in terms of the bin values $f_j$ to the wavelet coefficients $\tilde{f}_j$. If we do nothing other than transform from the $f_j$ to the $\tilde{f}_j$, the two descriptions are equivalent. The power in the wavelet analysis is in how the deviations are reflected in a subset of the wavelets, which simultaneously analyze different scales and can be filtered to enhance specific kinds of searches. For example, the oscillatory pattern largely lives in a set of wavelet coefficients of a single given level $\ell$.

**Fixed Resolution Global Significance:** From Eq.(6), it is clear that each bin of the distribution only contributes linearly to a single wavelet coefficient. If the individual bins are statistically independent, the wavelet coefficients for a single level are also statistically independent, allowing them to be trivially combined into a single statistical analysis.

A $p$ value can be calculated from Eq.(9) for each wavelet coefficient $\tilde{f}_{\ell,m}$, and translated into a test statistic $q_{\ell,m}$ defined as

$$q_{\ell,m} = -2 \ln p_{\ell,m}, \tag{11}$$

which obeys a $\chi^2$ distribution with two degrees of freedom. For wavelet coefficients of fixed $\ell$ the $q_{\ell,m}$ can be summed together to create a combined test statistic $q_\ell$,

$$q_\ell = \sum_{m=1}^{k} q_{\ell,m}. \tag{12}$$

If the $\tilde{f}_{\ell,m}$ are statistically independent then $q_\ell$ follows a $\chi^2$ distribution with $2k$ degrees of freedom, meaning that the statistical fluctuation in the ensemble of wavelet coefficients sharing the same $\ell$ can be easily quantified. In Eq.(20) in Appendix A we show that $p_\ell$, the combined $p$-value for all $\tilde{f}_{\ell,m}$ of a given $\ell$, can be written in terms of an incomplete gamma function.

This metric is highly useful for identifying features in the data that are spread over multiple coefficients within the same level of the wavelet transformation, and we refer to it as the fixed resolution global significance (FRGS). The situation is more subtle when an analysis requires combining multiple levels into a single statistical analysis, for example when searching for different local features of different scales.

## 3  Di-photon Mass Distribution

For a more realistic illustration we rely on a measured ATLAS di-photon invariant mass spectrum, $m_{\gamma\gamma}$ [21]. With its statistical fluctuations it allows us to perform a semi-realistic wavelet analysis with different injected signals. We choose the same patterns as in Sec. 2.2. After that we analyze the actual ATLAS results in a desperate attempt to search for new physics at the LHC.

### 3.1  Injected Signals

The background-only hypothesis for the ATLAS measurement shown in Fig. 3 is described by the functional form [20]

$$f_B(x) = N\,(1-x^{1/3})^b\,x^a, \qquad \text{with} \quad x = \frac{m_{\gamma\gamma}}{\sqrt{s}}. \tag{13}$$

We fit the coefficients $N$, $a$, and $b$ to the ATLAS di-photon spectrum [21], shown for reference in Fig. 3, and use this as a more realistic bases to inject the same four signal patterns used before, namely

1. a narrow Gaussian bump with mass 600 GeV and width 80 GeV;
2. a wide Gaussian bump with mass 750 GeV and width 300 GeV;
3. a bump-dip with a peak at 700 GeV and a dip 100 GeV below; and
4. an oscillation with a wave length of 265 GeV and a first peak at 415 GeV.

The combined kinematic distribution is binned into a histogram, subject to Poisson fluctuations. The injected signal pattern is normalized to give an approximately $5\sigma$ deviation in at least one of the wavelet coefficients.

The wavelet decompositions of the four resulting distributions are shown in Fig. 4. The top pane of each panel shows the resulting distribution in $m_{\gamma\gamma}$. The lowest six panes of each panel indicate the number of standard deviations in the corresponding wavelet coefficient compared to the background-only hypothesis, with color coding to guide the eye to more significant deviations. The second pane of each panel shows the reconstructed signal based on the indicated fraction of wavelet coefficients most significantly different from the background.

From Fig. 4, it is evident that both the narrow and wide resonant examples show the power of the wavelet transform to pick out the location and size of such a feature without making specific analysis choices beyond the initial binning of the histogram. Both are relatively well reconstructed with modest pixelation by a small fraction of 3% and 5% of the most significantly deviating wavelet coefficients. As in the toy example, the bump-dip is much more easily teased out by the wavelet that best matches its structure than a typical resonance search would be able to handle. In this case, a $5.5\sigma$ deviation in the $\ell = 3$, $m = 2$ wavelet coefficient correctly identifies its location and structure, and the reconstruction based on the 5% most significant wavelets reflects its structure. The oscillatory pattern is correctly identified at $\ell = 4$, where the

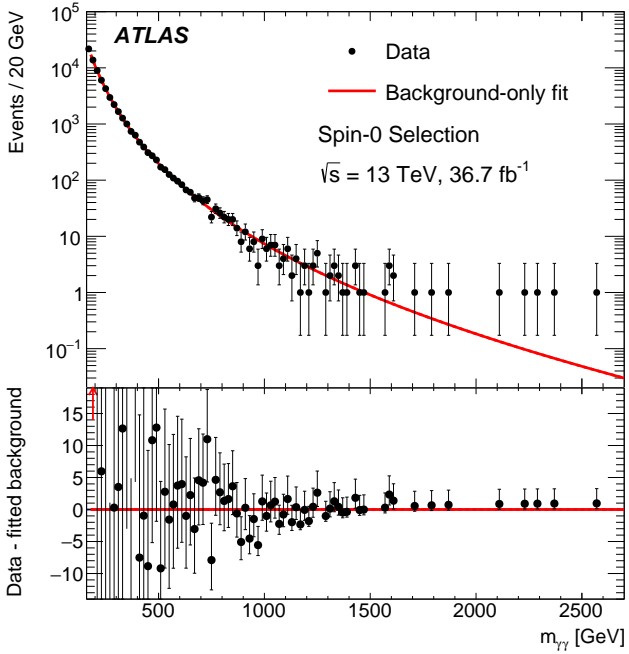

Figure 3: Di-photon invariant mass distribution after spin-0 resonance search selection from ATLAS [21] and background-only fit (upper panel). The lower panel shows the difference between data and the fit for each bin.

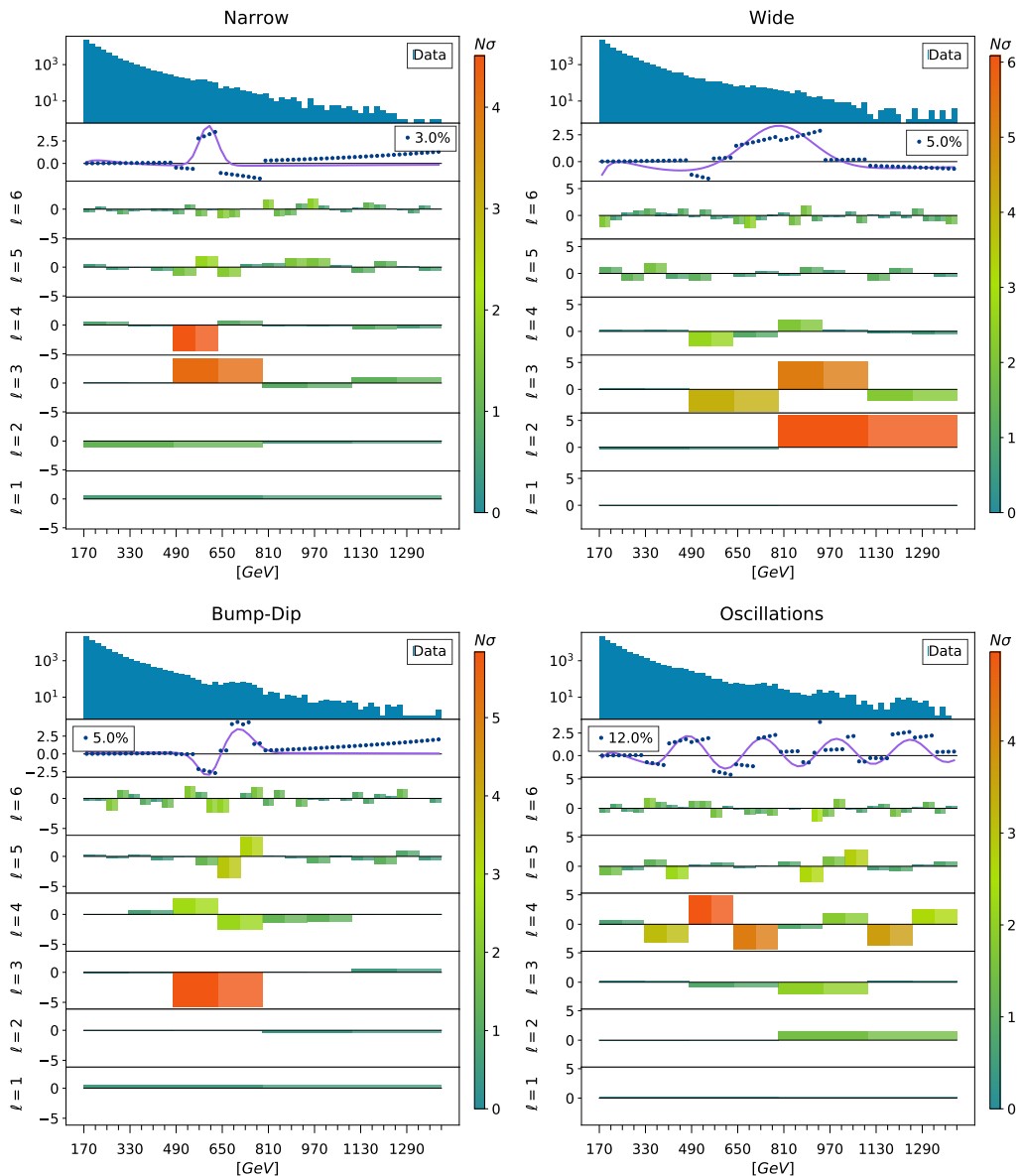

Figure 4: Wavelet transform of the di-photon invariant mass distribution with background hypothesis fit to the ATLAS data. We inject a narrow resonance (top left), wide resonance (top right), bump-dip (lower left), and oscillation pattern (lower right). The top panes show the input distribution, the next a signal reconstruction based on the indicated fraction of most significant coefficients, and the remaining panes the significance of each coefficient. The $x$-axis bins correspond to a linear scale between $m_{\gamma\gamma} = 200$ GeV and 1.45 TeV. In the second panel we show the signal function (in purple) that was used to generate the data for each function. Each wavelet coefficient is color-coded based on its deviation from the background hypothesis, with a color scale chosen individually for each plot based on the size of the most significant excess.

wavelet structure most closely matches the injected frequency. Its reconstruction in the second pane of the plot reflects the challenge of striking a balance between keeping enough coefficients to faithfully reconstruct the wave form, while excluding statistical noise and background.

As the reconstructed signal provides primarily qualitative information about the nature of

the statistical excess, there is no "correct" number of wavelet coefficients to use in the signal reconstruction. Instead of keeping a particular fraction of the coefficients, one could just as easily specify a minimum value of $N_\sigma$. Our choices in Fig. 4 to use 3%, 5% or 12% of the coefficients are roughly equivalent to setting $N_\sigma^{\min} \sim 2$. Without relying on this subjective benchmark, the presence or absence of new physics can be inferred directly from the analysis of individual wavelet coefficients, and from combined metrics like the fixed resolution global significance (FRGS).

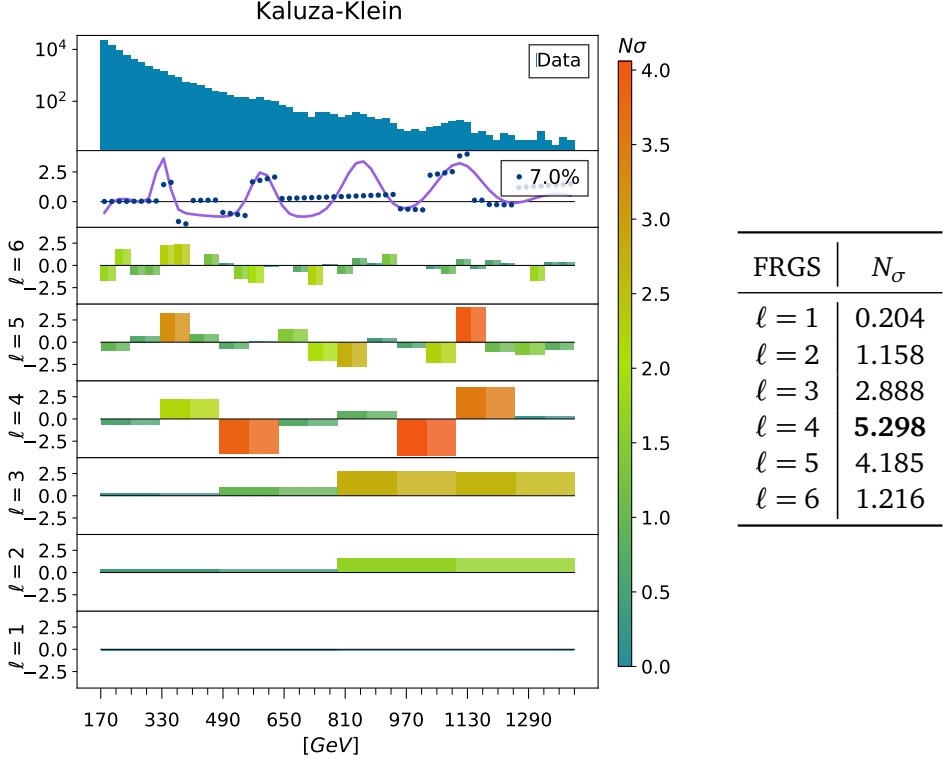

| FRGS | $N_\sigma$ |
|---|---|
| $\ell = 1$ | 0.204 |
| $\ell = 2$ | 1.158 |
| $\ell = 3$ | 2.888 |
| $\ell = 4$ | **5.298** |
| $\ell = 5$ | 4.185 |
| $\ell = 6$ | 1.216 |

Figure 5: The same as Figure 4, for the Kaluza Klein pattern described in the text. The table presents the FRGS for each level $\ell$. As with the oscillatory example from Figure 4, reconstructing the signal using (in this example) 7% of the wavelet coefficients involves a tradeoff between noise reduction and fidelity to the finer details of the injected signal. The FRGS, on the other hand, correctly identifies significant excesses in the $\ell = 4$, $\ell = 5$ and $\ell = 3$ resolution levels of $5.3\sigma$, $4.2\sigma$, and $2.9\sigma$, respectively.

A more realistic oscillatory pattern could correspond to a Kaluza Klein spectrum of resonances. We consider a series of resonances inspired by a warped extra dimension [24] for which the first resonance appears at $m_1 \approx 320$ GeV with a width of $\Gamma_1 \approx 18$ GeV, and subsequent masses and widths $m_i$ and $\Gamma_i$ are given by

$$m_i \approx \frac{x_i^{(1)}}{x_1^{(1)}} m_1, \qquad\qquad \Gamma_i \approx \frac{x_i^{(1)}}{x_1^{(1)}} \Gamma_1, \qquad (14)$$

where $x_i^{(1)}$ is the $i$th zero of the Bessel function $J_1(x)$.

This is a case where the signal is spread throughout the distribution, and the FRGS is useful to combine the significances from the statistically independent wavelet coefficients of a given level. In Fig. 5, we show the wavelet transform of this signal on top of the ATLAS background

model. Individual wavelet coefficients show up to $\sim 4\sigma$ deviations from the background model at $\ell = 3$ and $\ell = 4$, corresponding to the first three resonances in the tower. Combining the significances at each level, the FRGS indicates a $5.3\sigma$ deviation at $\ell = 3$, along with $3$–$4\sigma$ excesses at other resolutions.

This example illustrates the power of the wavelet transform and FRGS to tease out oscillatory signals, even when the 'frequency' of the signal is not constant. Our analysis could be just as easily applied to cases with large numbers of new states, for example [12] and [25], and to models with multiple resonances at arbitrary masses and widths.

## 3.2 ATLAS Distribution

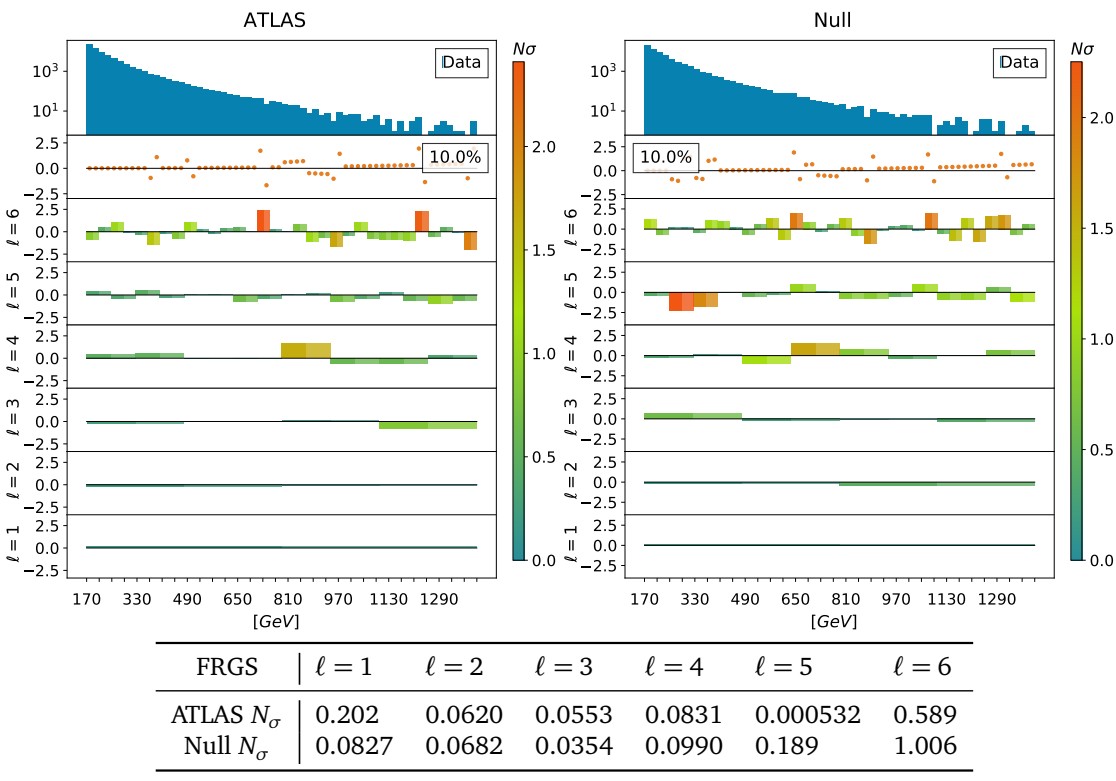

| FRGS | $\ell = 1$ | $\ell = 2$ | $\ell = 3$ | $\ell = 4$ | $\ell = 5$ | $\ell = 6$ |
|------|------------|------------|------------|------------|------------|------------|
| ATLAS $N_\sigma$ | 0.202 | 0.0620 | 0.0553 | 0.0831 | 0.000532 | 0.589 |
| Null $N_\sigma$ | 0.0827 | 0.0682 | 0.0354 | 0.0990 | 0.189 | 1.006 |

Figure 6: Top: wavelet analysis of the ATLAS $m_{\gamma\gamma}$ data (left) and an example null hypothesis distribution (right). Bottom: Fixed resolution global significance at each level for the ATLAS and 'Null' data sets.

Our final example is to analyze the actual ATLAS di-photon distribution [21], shown in Fig. 3. While we already know from the original analysis that it contains no indications of new physics, we can still use it as an example for our wavelet analysis tool in a realistic setting. The wavelet transform of the ATLAS di-photon data is shown in the left pane of Fig. 6. The fluctuations in all wavelet coefficients are small and reach the $2\sigma$ level in only two places. In the table below we give the FRGS at each level and, as expected, the ATLAS distribution indicates no signs of new physics. In fact, the wavelet coefficients appear to be slightly more consistent with the null hypothesis than one would naively expect. For instance, given the 64 bins translated into 32 coefficients at level $\ell = 6$ or 64 coefficients altogether we would expect around 20 to deviate at the $1\sigma$ level and 3 to deviate at the $2\sigma$ level.

We can compare the ATLAS result to a background-only set of toy data based on per-bin Poisson statistics, shown in the left pane of Fig. 6. Indeed, the statistical fluctuations are slightly more pronounced. From the corresponding Table we see that the difference is most

visible at the level $\ell = 5$. While it is beyond our ability to delve further in a meaningful way into what the origin of this feature is, one could imagine that it is the result of correlations between nearby $m_{\gamma\gamma}$ bins, which our analysis treats as independent. Correlations between bins and bin migration certainly have the potential to soften the statistical anomaly. In fact, one could imagine that the wavelet analysis might potentially offer a means to obtain interesting insights into such correlations in a way that is orthogonal to traditional approaches.

## 4 Outlook

Wavelets are a novel way to represent data in a way which, by simultaneously retaining information on multiple scales, allows for a flexible search for features on multiple scales. We have applied the Haar wavelet to a one-dimensional kinematic distribution, and demonstrated that local features of various sizes and global structures can both be disentangled. As toy examples we have shown how narrow and wide bumps, a bump-dip, and a KK-inspired oscillation pattern can be extracted from toy data as well as from an ATLAS di-photon mass spectrum. The background model is a simple, model-independent fit function.

We have discussed how the different features can be separated and understood from a universal analysis of wavelet coefficients, and how we can perform a statistical analysis on the wavelet coefficients. In the absence of correlations the translation from mass bins to wavelet coefficients is a simple linear transformation without any loss of information. Including correlations requires a proper statistical treatment. One of the most interesting aspects of our analysis is the fixed resolution global significance (FRGS) determined from one set of wavelet coefficients. To visualize the relevance of an anomaly we can also reconstruct the signal-background combination from the leading wavelet coefficients and find very good agreement with the injected signal. We hope that they will find fruitful use in future analysis of LHC data.

Our Kinematic Wavelet Analysis Kit (KWAK) is available as a numerical python package at https://github.com/alexxromero/kwak_wavelets.

## Acknowledgments

We acknowledge conversations with Daniel Whiteson, and inspiration from Carlos – who we hope will not disappoint us. The work of BGL and TMPT is supported in part by NSF Grant No. PHY-1620638. The work of BGL is also supported in part by the Chair's Dissertation Fellowship from the UCI Department of Physics & Astronomy. The work of AR is supported in part by NSF Grant No. PHY-1633631. This work was performed in part at the Aspen Center for Physics, which is supported by NSF grant PHY-1607611.

## A Statistical Method

Our statistical analysis is conducted on the coefficients of the Haar wavelet transformation of a binned distribution $f$, where $f_i$ is the number of events in the $i^{\text{th}}$ bin of the distribution. For this integer-valued signal we use a wavelet transformation with $\tilde{f}_{L,1} = f_1 - f_2$, $\tilde{f}_{L-1,1} = f_1 + f_2 - f_3 - f_4$, and so on, based on a basis of functions $h_{\ell,m}$ which are orthogonal but not normalized.

Given some hypothesis $H_0$ that predicts the mean expected value $\mu_i$ for each $f_i$ and under

the assumption of Poisson statistics, the probability distribution $P(\tilde{f}_{\ell,m}|H_0)$ can be shown to have the same form as Eq.(9). The derivation is simple, and relies on the observation that every $\tilde{f}$ can be written in the form $\tilde{f} = f_a - f_b$ for some Poisson-distributed variables $f_a$ and $f_b$. For wavelet coefficient $\tilde{f}_{\ell,m}$, these $f_{a,b}$ are given by

$$f_a = \sum_{j_{a,\,\mathrm{min}}}^{j_{a,\,\mathrm{max}}} f_j, \qquad j_{a,\,\mathrm{min}} = 2^{L-\ell+1}(m-1)+1 \qquad j_{a,\,\mathrm{max}} = 2^{L-\ell}(2m-1)$$

$$f_b = \sum_{j_{b,\,\mathrm{min}}}^{j_{b,\,\mathrm{max}}} f_j, \qquad j_{b,\,\mathrm{min}} = 2^{L-\ell}(2m-1)+1 \qquad j_{b,\,\mathrm{max}} = 2^{L-\ell+1}m. \tag{15}$$

As $f_{a,b}$ are both sums of Poisson-distributed variables, $f_a$ and $f_b$ follow Poisson distributions with mean values

$$\mu_{a,b} = \sum_{\mathrm{min}\, j_{a,b}}^{\mathrm{max}\, j_{a,b}} \mu_j, \tag{16}$$

and $P(\tilde{f}|H_0)$ is the Skellam distribution

$$P(\tilde{f}_{\ell,m} = \tilde{f}|H_0) = e^{-\mu_a - \mu_b} \left( \frac{\mu_a}{\mu_b} \right)^{\tilde{f}/2} \mathcal{I}_{\tilde{f}}(2\sqrt{\mu_a \mu_b}). \tag{17}$$

Signals of new physics may in general be manifested in the wavelet coefficients as positive or negative fluctuations in $\tilde{f}$ away from the mean expected value $\mu = \mu_a - \mu_b$, and so we use a two-tailed test to quantify the significance of a deviation. Given a background hypothesis $H_0$ and the measured value $\tilde{f}$ for each wavelet coefficient, we define the $p$-value as the likelihood of obtaining an outcome that is at least as extreme as the measured value, where by "more extreme" we mean "less probable". Expressed in terms of the finite sum over all $i$ such that $P(i|H_0) > P(\tilde{f}|H_0)$:

$$1 - p = \sum_{\forall i:\, P(i|H_0) > P(\tilde{f}|H_0)} P(i|H_0). \tag{18}$$

An excess can also be characterized by the number of standard deviations between $\tilde{f}$ and the mean expected value $\mu$, which in the Gaussian limit $\mu_a + \mu_b \gg 1$ is given by

$$N_\sigma = \sqrt{2}\, \mathrm{erf}^{-1}(1-p). \tag{19}$$

Even in the non-Gaussian limit of the Skellam distribution, it is often convenient to reference this definition of $N_\sigma(p)$ as a proxy for the $p$-value.

**Fixed Resolution Global Significance:** In a distribution with statistically independent bins, the wavelet coefficients within a given level $\ell$ are also mutually independent, making it straightforward to combine their significances. Following [26], the test statistic $q_i = -2\ln p_i$ obeys a $\chi^2$ distribution with two degrees of freedom: thus, the combined test statistic $q = q_1 + q_2 + \ldots + q_k$ with $k$ independent wavelet coefficients follows the $\chi^2$ distribution with $2k$ degrees of freedom, $\chi^2_{2k}$.

After computing $q_\ell = \sum q_m$ from all $m = 1, 2, \ldots, 2^{\ell-1}$ coefficients in the $\ell^{\mathrm{th}}$ level of the wavelet transformation, we calculate the fixed resolution global significance from the cumulative distribution function of the $\chi^2_{(2^\ell)}$ distribution:

$$D(\chi^2_{2k}) = \frac{\gamma\left(k, \frac{1}{2}\chi^2\right)}{\Gamma(k)} \quad \longrightarrow \quad p_\ell = 1 - \frac{\gamma\left(2^{\ell-1}, \frac{1}{2}q_\ell\right)}{\Gamma(2^{\ell-1})}, \tag{20}$$

where $\gamma(k, z)$ is the lower incomplete gamma function. This $p_\ell$ represents the likelihood that Poisson sampling of the hypothesis $H_0$ would return a value for the combined test statistic that is at least as large as $q_\ell$.

The fixed resolution global significance is particularly powerful for identifying signals that exhibit oscillatory behavior, whereas well localized signals such as simple bumps and bump-dips are more likely to be best identified by a small set of individual wavelet coefficients.

# B Kinematic Wavelet Analysis Kit

The Kinematic Wavelet Analysis Kit (KWAK) is a numerical Python package for the statistical analysis of binned distributions of a single kinematic variable. Its central function is to determine the probability distribution for each coefficient of the wavelet transformation of the data, and to identify the most significant deviations from a given background hypothesis. The KWAK package also provides a number of plotting options for displaying the results of the analysis, and is available online at https://github.com/alexxromero/kwak_wavelets, or installed via the command

```
pip install kwak
```

for either Python 2 or Python 3.

KWAK provides multiple options for calculating the probability distribution for each wavelet coefficient, including an exact approach based on Eq.(17), and three related approximate methods.

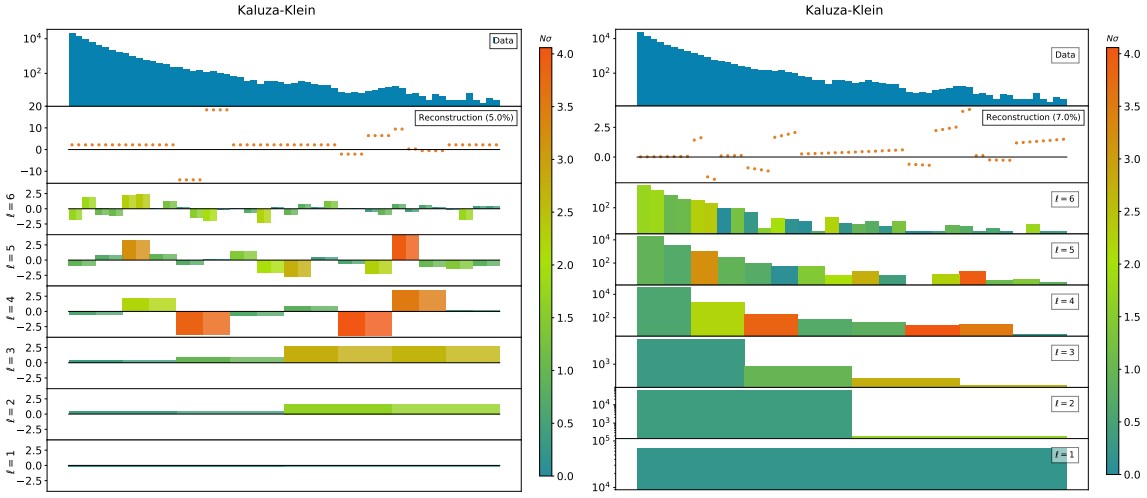

Figure 7: Left: the Kaluza-Klein model from the main text is used as a demonstration of the `nsigScalogram` plot with `reconstruction_scaled = nsigma_colorcode = False`. Right: a `wScalogram_nsig` plot of the same Kaluza-Klein model with `reconstruction_scaled = nsigma_colorcode = logscale = True` and `firsttrend = False`.

**Exact Method:** The exact approach is based on the assumption of Poisson statistics, and is valid specifically for kinematic distributions where the systematic error can be neglected. In this case the $p$-value for every coefficient in the wavelet transformation can be calculated by evaluating Eq.(18) directly, using the Skellam distribution of Eq.(17).

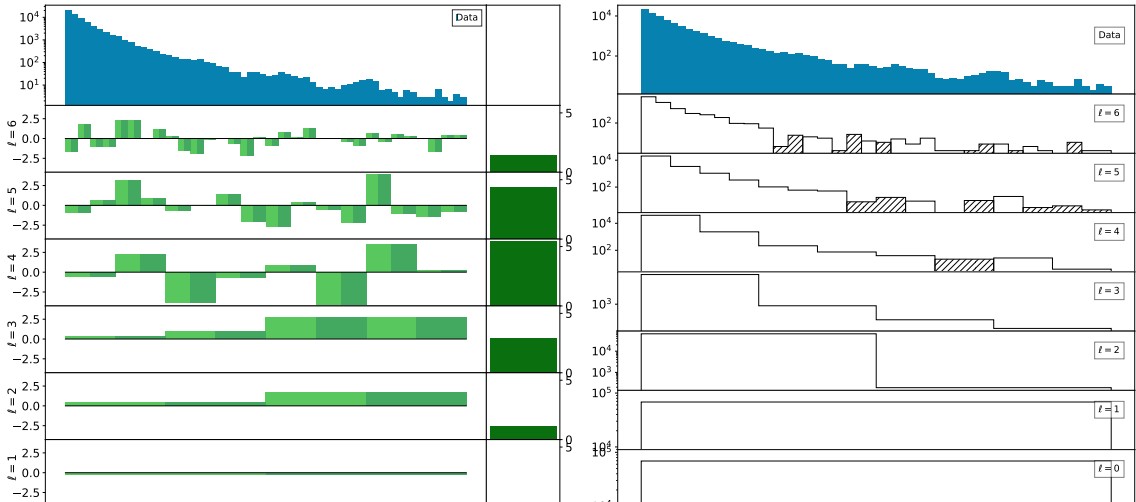

Figure 8: Using the same Kaluza-Klein model, two further plot examples are shown. Left: nsigFixedRes with nsigma_colorcode = False. Right: wScalogram with filled = False on a logarithmic scale with firsttrend = True.

This approach can be computationally intensive: the sums over less-extreme probabilities in Eq.(18) require repeated evaluation of the $k^{\text{th}}$ modified Bessel function of the first kind, where $k = \tilde{f}$ is an integer that scales with the number of events in the associated bins. Our KWAK implementation uses the mpmath Python library to conduct the calculation at arbitrary precision, to handle the exponentially large or small values of $\mathcal{I}_k(z)$. KWAK also uses mpmath to accommodate data sets with especially large fluctuations, where the individual probabilities $P(\tilde{f}|H_0)$ would otherwise be smaller than the floating point error.

These calculations are implemented in KWAK in the kwak.exact class:

```
kwak.exact(data, hypothesis, outputdir=None)
```

where data and hypothesis are one-dimensional arrays of equal length. If a value is provided for the optional keyword argument outputdir, the results of the analysis will be saved to a newly created directory with that name.

Instantiating the kwak.exact class creates several objects, including:

- self.Nsigma: the $p$-value for every wavelet coefficient, mapped to a value of "$N_\sigma$" following Eq.(19).

- self.NsigmaFixedRes: the fixed resolution global significance for each level of the wavelet transformation.

- self.Histogram: the probability distribution for each wavelet coefficient $P(i|H_0)$, calculated only for the values of $i$ necessary to evaluate the sum of Eq.(18).

Evaluating the 64-bin diphoton examples of Fig. 4 takes $\mathcal{O}(500)$ seconds when using the exact approach.

**Approximate Methods:** In situations where the precision of the exact method is unnecessary, or where the effect of systematic uncertainties cannot be neglected, it may be more appropriate to calculate $P(\tilde{f}|H_0)$ using one of the approximate methods of the kwak.nsets class. These three related approaches each approximate the wavelet coefficient probability distributions by

generating a large number, $N_{\text{sets}}$, of pseudo-random "data" sets drawn from the background-only hypothesis $H_0$ using Poisson statistics.[*] After performing a wavelet transformation on each pseudodata set, the `nsets` class assembles a histogram $D_{\ell,m}(\tilde{f}|H_0)$ for each wavelet coefficient, counting the number of pseudoexperiments $D_{\ell,m}$ which return a value $\tilde{f}_{\ell,m} = \tilde{f}$ for the $(\ell,m)$th wavelet coefficient. The probability distribution for that coefficient is approximated by:

$$P_{\ell,m}(\tilde{f}|H_0) = \frac{D_{\ell,m}(\tilde{f}|H_0)}{N_{\text{sets}}}, \tag{21}$$

where the histogram $D_{\ell,m}$ includes the values from $(N_{\text{sets}}-1)$ pseudoexperiments as well as the real data. Our choice to use an unnormalized wavelet transformation ensures that $\tilde{f} = \mu_1 - \mu_2$ is integer-valued.

This approach is limited by the fact that Eq.(21) does not resolve any probabilities smaller than $P_{\text{min}} = N_{\text{sets}}^{-1}$. Reliably distinguishing $4\sigma$ from $5\sigma$ deviations, for example, requires somewhat better than $N_{\text{sets}} = 10^7$, after accounting for the fact that there may be several values of $\tilde{f}$ for which $D(\tilde{f}|H_0) = 1$. Nevertheless, relatively small $N_{\text{sets}}$ can be sufficient for identifying deviations in the data, in much less time than is possible with `exact`. It also handles non-Gaussian distributions well: no assumptions about the shape of $P_{\ell,m}(\tilde{f}|H_0)$ are built in to this analysis.

The default implementation of the `nsets` method described above can be expanded with one of the two following options:

- `fastGaussian`: calculates the mean and standard deviation for each histogram $D_{\ell,m}$

- `extrapolate`: applies a functional fit to the histogram $D_{\ell,m}$, using an approximation of the Skellam distribution

With the first option, rather than defining the probability distribution $P_{\ell,m}$ and the $p$-value $p_{\ell,m}$, $N_\sigma$ is calculated directly and very simply from the mean $\mu(\tilde{f})$ and standard deviation $\sigma(\tilde{f})$ of the histogram $D_{\ell,m}$:

$$N_\sigma(\tilde{f}_{\ell,m}) = \frac{\tilde{f}_{\ell,m} - \mu(\tilde{f}_{\ell,m})}{\sigma(\tilde{f}_{\ell,m})}. \tag{22}$$

In the Gaussian limit of the Skellam distribution, $\mu_1 + \mu_2 \gg 1$, the `fastGaussian` approach provides a much better approximation of $N_\sigma$ for large fluctuations,

$$N_\sigma > \sqrt{2}\,\text{erf}^{-1}\left(1 - (\text{few}) \times N_{\text{sets}}^{-1}\right), \tag{23}$$

compared to what is possible with the default `nsets` method.

However, as seen in the left panel of Fig. 2, when $\mu_1 + \mu_2 < 1$ the Skellam distribution does not resemble a Gaussian at all, instead peaking sharply at $\tilde{f} = 0$. For rare processes with small but well-understood backgrounds, one or two events in some region of a kinematic distribution may be highly significant, requiring us to employ a better approximation of the Skellam distribution.

The `extrapolate` option is designed to handle both limits smoothly. It uses the curve fitter from `scipy.optimize` to fit the histograms $D_{\ell,m}$ with a modified Gaussian function

$$D_{\ell,m}(\tilde{f}) \approx n \exp\left(-\frac{1}{2}\left(\frac{\tilde{f} - \mu}{\sigma}\right)^2 - \gamma |\tilde{f}|^p\right) \tag{24}$$

for some $p \approx 1$ and $\gamma \geq 0$.

---

[*]Systematic effects could in principle be mimicked by adding some smearing to the Poisson mean $\mu_i$ in each bin of the pseudodata, but such modifications are left to the user.

Unlike the default version of `nsets` or the `fastGaussian` alternative, the `extrapolate` option requires a relatively large minimum value of $N_{\text{sets}}$ in order to run smoothly. If $N_{\text{sets}}$ is not large enough to generate nonzero entries in the histogram $D(\tilde{f})$ beyond the central values of $\tilde{f} = 0, \pm 1, \pm 2$, then the five parameter fit of Eq.(24) might not have a well-defined best fit point. For bins in the kinematic distribution with expected mean values $\mu_i \lesssim 10^{-1}$, it may be necessary to use $N_{\text{sets}} > 10^5$ to guarantee that `extrapolate` will provide a good fit for the probability distribution.

All three approximate methods are integrated into the `nsets` class:

```
kwak.nsets(data, hypothesis, nsets, seed=int, outputdir=None,
           fastGaussian=Boolean, extrapolate=Boolean)
```

where `nsets` $= N_{\text{sets}}$ determines the number of pseudoexperiments to generate, and `seed` specifies the seed to be used for the random number generator. By default, `fastGaussian` and `extrapolate` are set to False. Given conflicting inputs `fastGaussian = True` and `extrapolate = True`, the `fastGaussian = True` option takes precedence, and the `extrapolate` calculation will not be performed.

The `nsets` class also has `self.Nsigma`, `self.NsigmaFixedRes`, and `self.Histogram` objects; the only difference from the `exact` class is that for `nsets` the `self.Histogram` is the collection of histograms $D_{\ell,m}$, rather than the probability distributions $P_{\ell,m} = D_{\ell,m} \times N_{\text{sets}}^{-1}$.

**Comparison:** A rough guide to when (and when not) to use each of the four methods is given below:

- `exact`: Valid whenever the systematic uncertainties can be neglected. Especially useful at quantifying large fluctuations, and for cases where the evaluation time is not important.

- `nsets` (default): Provides fast analysis, best suited for data sets with moderate or small fluctuations. Valid for non-Gaussian probability distributions.

- `fastGaussian`: As fast as the default `nsets`, and able to distinguish between moderate and large fluctuations. Only valid for kinematic distributions where multiple events are expected in every bin.

- `extrapolate`: Expands the default `nsets` method to distinguish between moderate and large fluctuations, even in the non-Gaussian limit. Requires a larger minimum $N_{\text{sets}} \sim 10^5$ when operating in this limit.

As both the default `nsets` and the `fastGaussian` approximations can be run with $N_{\text{sets}} = 10^3 - 10^4$, these methods are the best choices if the analysis must be repeated many times.

The `fastGaussian` method remains accurate even for small values of $N_{\text{sets}}$: for example, calculating the FRGS for the Kaluza-Klein model shown in Fig. 5 with $N_{\text{sets}} = 10^3$ gives:

| KK FRGS ($N_\sigma$) | $\ell = 1$ | $\ell = 2$ | $\ell = 3$ | $\ell = 4$ | $\ell = 5$ | $\ell = 6$ |
|---:|:---:|:---:|:---:|:---:|:---:|:---:|
| exact: | 0.204 | 1.158 | 2.888 | 5.298 | 4.185 | 1.216 |
| nsets-default: | 0.422 | 0.850 | 2.422 | 3.022 | 2.959 | 0.893 |
| nsets-fastGaussian: | 0.230 | 1.157 | 2.859 | 5.267 | 4.459 | 1.497 |

Considering that `fastGaussian` with $N_{\text{sets}} = 10^3$ already approaches the accuracy of the `exact` method, and evaluates almost 1000 times more quickly, there is a real benefit to taking the Gaussian approximation if appropriate.

In the Gaussian limit with multiple events expected in every bin, the `extrapolate` approach can be used with a smaller minimum $N_{\text{sets}} \ll 10^5$. Below $N_{\text{sets}} < 10^4$, the evaluation time becomes dominated by the curve fitting function, so that $N_{\text{sets}} = 10^3$ takes as long to evaluate as $N_{\text{sets}} = 10^4$. Thus, the primary purpose of `extrapolate` is to provide improved accuracy in the $10^4 < N_{\text{sets}} < 10^6$ range, especially for cases when the Gaussian approximation is not necessarily appropriate.

Around $N_{\text{sets}} = 1.5 \times 10^6$, the three approximate calculations and the `exact` method take equivalent amounts of time to evaluate. Unless systematic uncertainties are being included in the calculation, there is no benefit to running any of the `nsets` approximations with $N_{\text{sets}} > 10^6$, as `exact` becomes faster at this point.

**Plotting Functions and Options:** The plots of Figures 4, 5, and 6 are generated using one of the plot types included in the KWAK package, `kwak.nsigScalogram`:

> `kwak.nsigScalogram(data, hypothesis, nsigma, *kwargs)`

where `nsigma` should be the `self.Nsigma` object from an `exact` or `nsets` class. The top two panels of this plot show a histogram of the data, and a reconstruction of the putative signal using only the wavelet coefficients with the largest deviations away from the background hypothesis. The remaining panels show the value of $N_\sigma$ for each wavelet coefficient.

In addition to the mandatory arguments, a number of optional keyword arguments can be used to change characteristics of the plot:

- For the reconstruction of the signal:

    - `nsigma_min = x`: Uses only wavelet coefficients with $N_\sigma > x$.
    - `nsigma_percent = x`: Uses only the most significant $x \times 100\%$ wavelet coefficients.
    - `reconstruction_scaled = Boolean`: Provides an option to divide all of the entries in the reconstructed signal by the square root of the mean expected value for that bin, so that the $y$ axis corresponds loosely to "$N_\sigma$" rather than the number of events in the signal.

- `nsigma_colorcode = Boolean`: Color codes the plot of the wavelet coefficients with a scheme based on the size of $N_\sigma$.

- `title = str`: Prints a title above the plot, in size 18 font.

- `xlabel = str`: Prints a label for the $x$ axis, in size 14 font.

- `outputfile = str`: Saves the plot as a PNG file with name `"outputfile"`.

As an example of the default output of `nsigScalogram`, Fig. 7 shows the Kaluza-Klein model of Fig. 5 but with `reconstruction_scaled = nsigma_colorcode = False`.

Rather than plotting $N_\sigma$ for each wavelet coefficient, the plotting function `kwak.wScalogram_nsig` replaces $N_\sigma$ with the values of the wavelet coefficients themselves. In addition to the keyword arguments available for `nsigScalogram`, `kwak.wScalogram_nsig` has an option to plot the values of the wavelet coefficients on a logarithmic scale:

- `logscale = Boolean`.

Negatively signed wavelet coefficients are shown as positive values with hatched lines on the logarithmic plot, as shown in the right panel of Fig. 7. A second additional optional argument, `firsttrend = Boolean`, determines whether or not the value of the $\tilde{f}_{\ell=0}$ coefficient is shown.

In the plots of the main text, the FRGS is typically shown as a separate table. Another plotting method, `kwak.nsigFixedRes`, shows the FRGS $N_\sigma$ value as an additional column on the right:

```
kwak.nsigFixedRes(data, hypothesis, nsigma, nsigma_FRGS, *kwargs)
```

also with the optional keyword arguments corresponding to color-coding and plot labels. An example with the default color coding is shown in the left panel of Fig. 8.

Finally, to display the wavelet transformation of the data without any reference to the statistical analysis, we provide

```
kwak.wScalogram(data, *kwargs)
```

- `logscale` = *Boolean*
- `firsttrend` = *Boolean*
- `filled` = *Boolean*
- `outputdir` = *str*

where the new optional argument `filled` determines whether or not to fill the histograms for the wavelet coefficients with a solid color. As before, negative coefficients on the logarithmic scale are shaded with hatch marks. An example with `filled = False` is shown in the right panel of Fig. 8.

For additional control over the relative sizes of the individual panels in each plot, the range of $y$ values shown for a particular panel, the text displayed inside the legends, or other similar details, the user can edit the relevant parameters directly in `nsigmaplots.py` and `scalograms.py` in the `kwak/plotting` folder.

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
