# Peer review of "Multi-scale Mining of Kinematic Distributions with Wavelets"

_SciPost Physics, doi:SciPost Phys. 8, 043 (2020)_

## Round 2 · Referee Report · Anonymous · 2019-9-5

Strengths

1- interesting and useful idea
2- wavelets seem more naturally and easily suited to this use than to more complicated uses in the existing literature (jet substructure, pile up, etc)
3- great to make these tools available more widely!

Weaknesses

1- exposition is not very clear (lots of definitions in a somewhat rapid-fire manner, and then some sloppiness when "plugging in" certain definitions)
2- some choices seem to be arbitrary

Report

In general, this is a good idea applied to what seems like a natural use case. I think there are a few specific choices to expand on and justify and a few small corrections, but over all I think it's a very good introductory / exploratory study.

Requested changes

1- the authors should show the signal that has been injected in the top panels of Fig 1 again in the second panel (with proper normalization) so that we can see how well their "10% leading" prescription does -- I think this is pretty important, since otherwise we don't know if this technique actually does recover the signal. Perhaps this is a functionality that needs to be built into KWAK. (Incidentally, I also don't see \tilde f_0, which they say is shown.)
2- I'm confused by Eq 6 (which I think is of central importance -- so confusion here is not good!). What is the object f that has no tilde but has a (single) subscript? How is this related to the (well defined) object \tilde f_{l,m} or to f?
3- I confirm Eq 9, but again I'm confused by what \tilde f means here and how it relates to these f's that have single subscripts. I think the subscripts here are different than the ones in point 2, but this is very confusing...
4- "valorous" or "quixotic" are better than "desperate" in the beginning of Sec 3 :)
5- They should explain or justify their "color coding" in Fig 4 and after -- "how significant" is a coefficient to be shown in red?
6- Another key point: how are 3%, 5%, 12% chosen in Fig 4? How sensitive are their results to these choices? And again in Fig 4, it would be great to the see the injected signal as well. (Minor point about Fig 4, they say it extends to 2.6 TeV but appears only to go to ~ 1.5 TeV)
7- When they discuss Kaluza-Klein models, can they comment on other newer models with many particles? e.g. 1902.05535, or their current ref 6
8- The penultimate sentence of Sec 3.2 appears to be missing a word or words
9- In Eq 15, is the lower case p supposed to be P? If not, how are they related?
10- The second part of their Appendix A about the FRGS should be promoted to a section or at least a subsection of Sec 2. This is very interesting and (seemingly) allows for a statistical test of their wavelet analysis without any of the arbitrary numbers like 3%, 5% etc above. Removing such model-dependence is very useful with a new analysis procedure like the one they propose

---

## Round 3 · Referee Report · Anonymous · 2020-2-7

Report

The new draft is noticeably improved, easier to read, and more useful as a reference.

---

## Round 3 · Author Response

In our resubmitted manuscript, we have made a few changes to the text, most notably to emphasize the utility of the fixed resolution global significance (FRGS) as a model-independent analysis tool. We have also made modifications to the text and figures to address typos and to add clarity to certain sections.

---

## Round 3 · List of Changes

1. In Fig.1, Fig.4 and Fig.5 we have added the original injected signal in the second panel of each plot.

2. In Section 2.1 we have added text to clarify that the discrete signal "f_j" and the function "f(x)" represent the same distribution.

3. We have added a paragraph in Sec. 2.3. to introduce the fixed resolution global significance (FRGS) in the body of the paper.

4. In Section 3.1 on page 10 we add a paragraph describing how the fraction of wavelet coefficients to use in the signal reconstruction in Fig.4 provides primarily a qualitative description of the excess signal, and that the choice to use 3%, 5%, 10% or some other fraction does not affect the statistical analysis.

Resubmission 1906.10890v3 on 5 February 2020
Submission 1906.10890v2 on 28 August 2019

---

## Editorial Decision

published